# CROSS RESOLUTION ENCODING-DECODING FOR DETECTION TRANSFORMERS

## ABSTRACT

Detection Transformers (DETR) are renowned object detection pipelines, however computationally efficient multiscale detection using DETR is still challenging. In this paper, we propose a Cross-Resolution Encoding-Decoding (CRED) mechanism that allows DETR to achieve the accuracy of high-resolution detection while having the speed of low-resolution detection. CRED is based on two modules; Cross Resolution Attention Module (CRAM) and One Step Multiscale Attention (OSMA). CRAM is designed to transfer the knowledge of low-resolution encoder output to a high-resolution feature. While OSMA is designed to fuse multiscale features in a single step and produce a feature map of a desired resolution enriched with multiscale information. When used in prominent DETR methods, CRED delivers accuracy similar to the high-resolution DETR counterpart in roughly 50% fewer FLOPs. Specifically, state-of-the-art DN-DETR, when used with CRED (calling CRED-DETR), becomes 76% faster, with $\sim 50\%$ reduced FLOPs than its high-resolution counterpart with 202G FLOPs on MS-COCO benchmark. We plan to release pretrained CRED-DETRs for use by the community.

## 1 INTRODUCTION

Detection Transformers (DETR) (Carion et al., 2020) are end-to-end object detection frameworks without post-processing, such as anchor boxes, box matching, and non-maximal suppression (NMS) that are required in ConvNet-based detectors (Ren et al., 2015; Liu et al., 2016). Since their first entry (Carion et al., 2020), DETRs have evolved, mainly regarding query design (Gao et al., 2021b; Zhu et al., 2021; Meng et al., 2021; Wang et al., 2021; Liu et al., 2022; Li et al., 2022) to improve the detection performance, while recent DETR pipelines achieve that via salient points (Liu et al., 2023) or unsupervised pretraining (Chen et al., 2023).

State-of-the-art DETRs exploit high-resolution features (known as Dilated Convolution or DC variant) (Liu et al., 2022; Li et al., 2022) or dense multi-scale features (Zhang et al., 2023b) to push the detection accuracy further. However, they suffer from high computation complexity w.r.t. their accuracy gains. For example, DAB DETR (Liu et al., 2022) with high-resolution features improved 2.3AP, but it introduces a 114% rise in FLOPs.

The primary reason is the quadratic computational complexity (Dai et al., 2021) of the Transformer's attention mechanism (Zhang et al., 2023a) w.r.t. the spatial size, i.e., a Transformer is $O(H^2W^2)$ complex in processing a feature map of a spatial size of $H \times W$. Hence, doubling the resolution of the encoder input quadruples its computations while also affecting the decoder. Although deformable attention (Zhu et al., 2021) addresses this issue, it causes additional runtime overhead due to irregular memory accesses.

To address this issue, recent IMFA (Zhang et al., 2023a) proposes using a low-resolution feature map and Top-K sparsely sampled higher-resolution features (Figure 1). IMFA exhibits improvements in accuracy with lower FLOPs. However, sparse sampling incurs memory access costs due to irregular memory access, which becomes prevalent with more samples. Recent (Li et al., 2023) exploits attention among only interleaved tokens for reducing computations in multiscale attention in the encoder. (Zhao et al., 2024b) works on reduced resolution images ($640 \times 640$) instead of high resolution settings ($800 \times 1200$) on standard DETR.

Our aim aligns with improving DETR speed and accuracy by using multiscale features at high-resolution settings. However, our key motivation is based on our finding that the encoder consists

Figure 1: Left: Single-Scale and/ DC DETR. Middle: IMFA DETR (Zhang et al., 2023a). Right: CRED DETR (Ours). Multiple arrows between two modules indicate layerwise refinement. Stage-1 features are generally not used due to large resolution and small receptive field.

of most computations relative to the decoder. Therefore, we propose a principal design change in the DETR pipeline, i.e., feeding the encoder with low-resolution while feeding the decoder with a high-resolution feature obtained from the backbone. By keeping the encoder input low-resolution, we save computations, while by keeping the decoder input high-resolution, we provide the decoder access to fine-grained details. Since the high-resolution features from the backbone lack large spatial context (Carion et al., 2020), we develop a novel Cross Resolution Attention Module (CRAM). It utilizes encoder output that has a global context and transfers this information into the high-resolution feature map. When this feature map is fed to the decoder, the decoder has access to the fine-grained details and the global context, thus improving the accuracy and runtime.

Then, by exploiting this capability of CRAM to transfer the information from low-resolution encoder output to high-resolution feature, we propose to reduce the resolution of an encoder further to save computations. This behavior is intended to develop faster DETRs offering speed-accuracy tradeoffs. However, feeding the encoder naively with reduced resolution degrades its performance. Hence, we devise a novel module called One Step Multiscale Attention (OSMA), which attends to multiscale information in one step and can produce tokens or feature maps of the desired resolution enriched with multiscale information. When the encoder is fed the tokens produced by OSMA at aggressively low resolution, accuracy is considerably improved while avoiding any runtime overhead relative to the baseline which was fed with a low-resolution backbone output.

We name our overall approach as Cross-Resolution Encoding-Decoding (CRED), shown in Figure 1. We demonstrate that our CRED-enhanced DETR can attain an AP (average precision) equivalent to the original DETR's high-resolution counterpart in 50% fewer FLOPs at 76% improved runtime. For instance, DN-DETR (Li et al., 2022)+CRED reduces FLOPs from 202G to 103G ($\sim 50\%\downarrow$) and improves FPS from 13FPS to 23FPS ($\sim 76\%\uparrow$) without losing accuracy. In addition, applying CRED in DETR variants (Meng et al., 2021; Liu et al., 2022; Zhang et al., 2023a) consistently improves their accuracy and runtime compared with their DC variants. To improve runtime further, we half the encoder resolution via OSMA. Interestingly, we only observe $-1$AP in this configuration while the runtime is further improved by 84% compared with the vanilla DC variant. This signifies the potential of Cross-Resolution Encoding-Decoding mechanism in DETRs.

## 2 PRELIMINARY

This section revisits the architectural design of vanilla DETR (Carion et al., 2020) and advanced DETRs (Liu et al., 2022; Li et al., 2022; Zhu et al., 2021). DETR comprises a backbone, a Transformer encoder, and a Transformer decoder. In a backbone (Li et al., 2022), it is common to keep five stages, each operating at a resolution half of its previous stage. Thus, the final stage (stage-5) runs at a *stride of 32* at the original resolution. Once an image $I \in \mathbb{R}^{3 \times H \times W}$ is fed into a backbone (Figure 1), the backbone output $F_b \in \mathbb{R}^{C \times H_0 \times W_0}$ is fed to a Transformer encoder, producing encoded feature embeddings or tokens $F_e \in \mathbb{R}^{d \times H_0 \times W_0}$. These embeddings are fed to the Transformer decoder to produce a fixed number of queries ($N_q$), each representing an object detectable in the image. The queries are passed through two Feedforward neural Networks (FFN) to obtain the object class and its bounding box. During training, bipartite or Hungarian matching (Carion et al., 2020) is performed for one-to-one assignments of ground truth and predictions.

In DETR pipelines (Carion et al., 2020; Liu et al., 2022; Li et al., 2022), the embeddings $F_e$ are directly fed to the decoder where the $N_q$ queries interact with $H_0 \times W_0$ features in $F_e$ via cross attention.

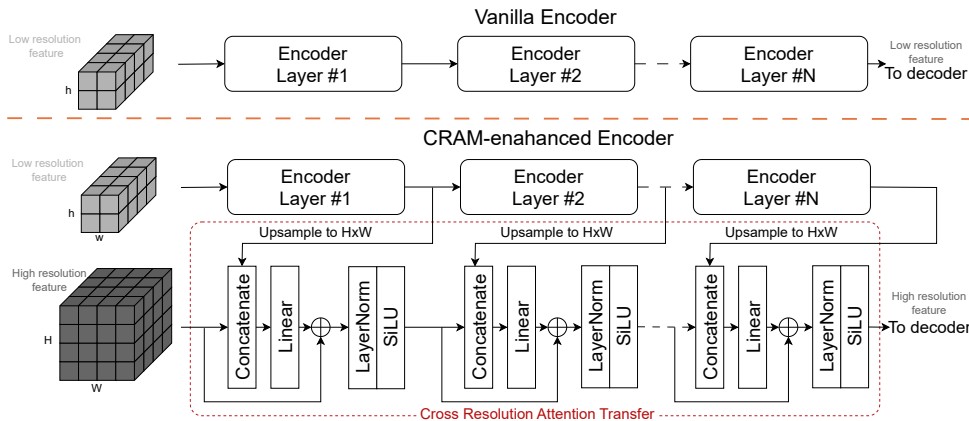

Figure 2: CRAM: Cross Resolution Attention Module.

The encoder complexity is $O(H_0^2 W_0^2)$ in its self-attention, whereas for the decoder, it is $O(N_q H_0 W_0)$ in cross attention and $O(N_q^2)$ in its self-attention. In DETRs, $N_q$ is relatively smaller, i.e., 300; however, $H_0 W_0$ is quite a large number depending on the image size, e.g., $1280 \times 800$ (Liu et al., 2022; Li et al., 2022). Hence, *most of the computations are concentrated in the encoder.* When opting for high-resolution detection, known as the DC variant (Liu et al., 2022; Li et al., 2022), the backbone output stride is set lower than 32. For this purpose, stage-5 of the backbone is set to run at *stride 16* (Meng et al., 2021; Liu et al., 2022; Li et al., 2022) w.r.t. the image, thereby doubling the resolution of $F_b$. This leads to quadratic growth in the encoder computations due to the increased resolution (Dai et al., 2021).

In this paper, we rethink the encoder-decoder information flow while leveraging multiscale features in a computationally efficient manner. Summarily, we propose an approach that feeds the encoder with a stride $\geq 32$ while feeding the decoder with a stride $\leq 32$ to harness the best of both worlds, i.e., the accuracy of high-resolution detection and speed of low-resolution detection. To our knowledge, such a mechanism has not been demonstrated in DETRs.

## 3 METHOD

We propose to enhance the DETR design with the two modules. *Firstly*, we develop CRAM which enables Cross-Resolution Encoding-Decoding. *Secondly*, we develop OSMA module, which facilitates generating feature maps enriched with multiscale information in a computationally efficient manner and further enhances the performance of CRED.

### 3.1 CROSS RESOLUTION ATTENTION MODULE (CRAM)

As mentioned in Sec. 2, high-resolution detection improves DETR accuracy, but high-resolution input to the encoder is compute-intensive due to its quadratic complexity (Dai et al., 2021). Hence, we feed the encoder with low resolution while feeding the decoder with high resolution, calling it *cross-resolution encoding-decoding paradigm.* However, the information flow between them no longer exists due to the different input sources to the encoder and decoder. Hence, we develop CRAM that acts as a bridge between the encoder and decoder in the proposed cross-resolution encoding decoding paradigm. The overall design of our approach (CRAM) is shown in Figure 2.

Consider two feature maps $X \in \mathbb{R}^{C \times h \times w}$ and $Y \in \mathbb{R}^{C \times H \times W}$. In our Cross-Resolution Encoding-Decoding approach, the low-resolution feature $X$ is fed to the encoder layers, while the high-resolution feature $Y$ is fed to the attention transfer module. To transfer the knowledge of the encoder embeddings ($X_e = \text{Encoder}(X)$, $X_e \in \mathbb{R}^{C \times h \times w}$) to $Y$, we spatially upsample ($\hat{X}_e = \text{Upsample}(X_e)|_{H,W}$ $\hat{X}_e \in \mathbb{R}^{C \times H \times W}$) the encoder output to match the resolution of $Y$, and a concatenation operation is performed.

Now, we use a linear projection layer, which combines the features information of the high-resolution features and the upsampled low-resolution feature from the encoder. The output ($Z = \text{Linear}(\text{Cat}(Y, X_e))$) is normalized via LayerNorm (Ba et al., 2016) and passed through SiLU

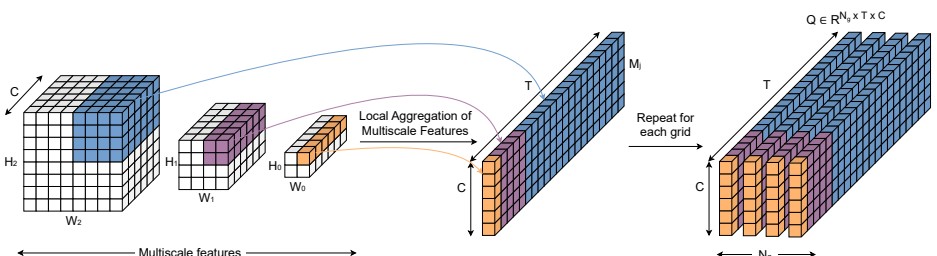

Figure 3: Local aggregation of the multiscale features in OSMA for $g_0 = 1$ which produces $Q \in \mathbb{R}^{N_g \times T \times C}$

activation (Hendrycks & Gimpel, 2016) ($\hat{Y}_e = \text{SiLU}(\text{LayerNorm}(Z))$, $\hat{Y}_e \in \mathbb{R}^{C \times H \times W}$). We use the residual connections to facilitate smoother optimization. This process is repeated as many times as the number of encoder layers to refine the high-resolution $Y$ by transferring the knowledge embedded in the encoder layers. The input to CRAM is initialized with the high-resolution feature from an intermediate backbone stage (Sec. 2), whereas the final $\hat{Y}_e$ is fed to the decoder for the predictions

**Computation Complexity**  An encoder layer has $\mathcal{O}(h^2w^2)$ complexity to process its input $X_e \in \mathbb{R}^{h \times w}$. In our case, we feed the encoder with $X_e \in \mathbb{R}^{H/r \times W/r}$, where $r \geq 2$, and feed the decoder with $X_d \in \mathbb{R}^{H \times W}$. In aggregation, we have a total complexity of $\mathcal{O}(\frac{H^2W^2}{r^4} + HW)$, which is far lower than the vanilla encoder complexity of $\mathcal{O}(H^2W^2)$. With this strategy, we achieve accuracy equivalent to when the encoder is fed with the high-resolution feature while having at least 50% fewer overall computations for $r = 2$. Specifically, w.r.t. vanilla encoder with high resolution, a FLOP saving of 50% is obtained. See Sec 4.2 for the computational complexity analysis.

**Why Could Cross Resolution Attention Transfer Improve Performance?**  In vanilla DETR design, the encoder embeddings are produced from stage-5 (low-resolution) and have global receptive filed via self-attention (Dosovitskiy et al., 2020), which are fed to the decoder for the predictions. However, these embeddings do not have fine details of smaller objects. On the other hand, the high-resolution input $Y$ (earlier stages, e.g., stage-4) to CRAM has a small receptive field but has details of smaller objects.

In CRAM, the concatenation operation followed by a linear layer infuses the local and global context to produce fine-grained, high-resolution features, similar to what earlier semantic segmentation approaches (Zhao et al., 2017) used for improving performance. In the same way, with this operation, the high-resolution feature $Y$ acquires a global receptive field when concatenated with the encoder embedding or tokens $X_e$. After the layerwise refinement, it is fed to the decoder, which improves the accuracy and speed; even the encoder still functions at a smaller resolution.

## 3.2 ONE STEP MULTISCALE ATTENTION (OSMA)

In single-scale operation of DETRs (Liu et al., 2022; Li et al., 2022), the backbone output $F_b$ is fed to the encoder. However, $F_b$ does not have direct access to the multiscale information. Whereas multiscale DETRs feed the encoder with either sampled (Zhu et al., 2021) or dense multiscale features (Zhang et al., 2023b), which is computationally heavy.

We propose OSMA, which produces $F_o$ enriched with multiscale information, i.e., the best of both single-scale and multiscale methods without progressive fusion, i.e., fusing two scales at a time (yol)(Figure 1). In general, $F_b$ has a stride 32 w.r.t. the input (He et al., 2016); however, using multiscale features, OSMA can produce a feature map $F_o$ ('o' refers to OSMA) of stride greater or lesser than 32 which directly controls the encoder computations. With this functionality, OSMA offers better features or tokens enriched with multiscale information to be utilized by the encoder.

OSMA has three main steps: *First*, Local aggregation of multiscale features, *Second*, performing one-step attention on the aggregated features, and *Third*, broadcasting the output into desired resolution.

**Local Aggregation of Multiscale Features.** This step aggregates $n$ multiscale features $F^i \in \mathbb{R}^{C \times H_i \times W_i}$ obtained from the backbone, where $i = 0, 1, ...n-1$ is the scale index (Figure 3). In this step, all the

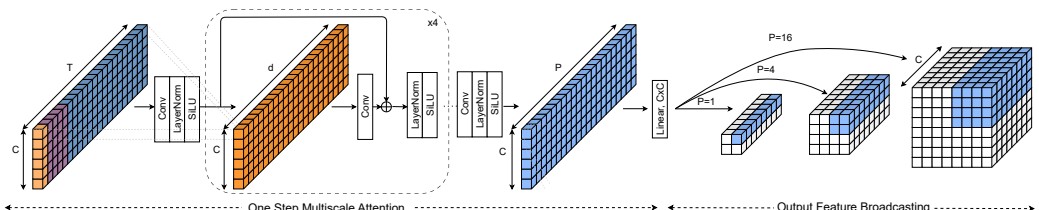

Figure 4: One step attention and Output Broadcasting in OSMA. 'One-Step' should not be confused with 'single layer'; instead, it refers to all the multiscale features being attended simultaneously through $1 \times 1$ layer. Output broadcasting infers the shape of the output feature based on the value of $P$.

features are first divided into non-overlapping grids of size $g_i = 2^i * g$. The total number of grids for all the scales is equal and is given by $N_g = H_i/g_i \times W_i/g_i$.

Let $F_j^i \in \mathbb{R}^{C \times g \times g}$ denotes feature map of $j^{th}$-grid of $F^i$, where $j = \{0, 1, ... N_g\}$. For each scale, we flatten $F_j^i$ in spatial dimension and stack all its features with its corresponding grids in other scales, as shown in colors in Figure 3. This results in a matrix $M_j \in \mathbb{R}^{C \times T}$ for each grid. This step is repeated for each non-overlapping grid, and we obtain $N_g$ matrices, referring as $Q \in \mathbb{R}^{N_g \times T \times C}$. The number $N_g$ is absorbed into the batch dimension while processing batched training or inference so that all the matrices can be processed in parallel. This step requires all the feature maps $F^i$ to be an integral multiple of $g$. Hence, we align all the feature maps w.r.t. $g_i$ before performing local aggregation via bilinear interpolation. See Table 5 for ablation on '$g$'.

**One Step Multiscale Attention.** We perform information fusion by attending to all the multiscale features and their channels simultaneously, thus calling it a one-step multiscale attention. We describe this process for a single grid or $M_j$, also depicted in Figure 4.

In the attention process, $M_j$ undergoes a $1 \times 1$ convolution operation with weights $\mathbf{W} \in \mathbb{R}^{d \times T \times 1 \times 1}$ which projects $M_j$ in a $d$ dimensional latent space. The $1 \times 1$ layer combines information from all the $T$ multiscale features with a unit stride. In this way, the output of multiscale information becomes richer. Then, we perform a layer normalization over the columns of $M_j$ followed by SiLU activation (See Sec. 4.2).

This process is repeated for feature refinement, and the penultimate $1 \times 1$ layer produces $P$ channels. The final layer is a linear projection applied over the columns of $\hat{M}_j$ to refine the column information because a column becomes a feature $\in \mathbb{R}^C$ in the output feature map $F_o$. The output of this step $\hat{Q} \in \mathbb{R}^{N_g \times P \times C}$ is broadcasted into a feature based on the requirements, as discussed next. **Output Feature Broadcasting.** This step broadcasts $\hat{Q}$ into a feature map $F_o$ of the desired resolution based on the pair $\{g_0, P\}$. For example, for $g_0 = 1$, if we aim to produce a feature $F_o$ of size $H_0 \times W_0$, the value of $P$ will be set to 1, or if a feature $F_o$ of size $2H_0 \times 2W_0$ is required, $P$ can be set to 4, or if a feature $F_o$ of size $H_0/2 \times W_0/2$ is needed, $\{g_0 = 2, P = 1\}$ can be used.

This flexibility of generating $F_o$ of desired resolution allows controlling the encoder's input resolution and, hence, its computations. Meanwhile, multiscale information infusion helps improve DETR accuracy with a slight computational overhead. We have studied various $\{g_0, P\}$ combinations, and our empirical results show that $\{g_0 = 1, P = 1\}$ is the best combination for keeping the resolution of $F_o$ equals to $F_0$ whereas $\{g_0 = 2, P = 1\}$ is best for reducing the resolution. See Table 5.

### 3.3 CONFIGURING CRED FOR DETRs

We mainly test two important configurations. In all the configurations, OSMA feeds the encoder, whereas CRAM feeds the decoder. OSMA is always fed with $\{F^3, F^4, F^5\}$ output of backbone (Figure 1). Each configuration has different settings for OSMA and input sources for CRAM.

**Default.** OSMA operates at $\{g_0 = 1, P = 1\}$ i.e. its output resolution is same as $F^5$. Whereas CRAM is fed with $F^4$. In other words, the encoder runs at half the resolution of the decoder.

**Configuration: 'DC$\times$0.25'.** OSMA operates at $\{g_0 = 2, P = 1\}$, producing feature map of half the resolution of $F^5$ or quarter of $F^4$ or quarter of DC. Whereas CRAM is fed with $F^4$. This configuration

evaluates the capability of our Cross-Resolution Encoding-Decoding when the encoder is fed with a very low-resolution map. For comparison, the encoder in baselines is fed with $F^5$ downsampled to half of its resolution. Since this resolution is $1/4$ of DC resolution, it is named as DC$\times$0.25.

**Additional Configuration: 'OO'** This is similar to the default configuration except the input source to CRAM. We use two instances of OSMA: $OSMA_e$ and $OSMA_c$. The former operates at $\{g_0 = 1, P = 1\}$, feeding the encoder, while the latter operates at $\{g_0 = 1, P = 4\}$, i.e. in upsampling mode, feeding CRAM with a resolution equivalent to $F^4$. This configuration tests the capability of OSMA to fuse multiscale information while increasing the output resolution.

Based on the computational budget, $F^3$ can also be fed to CRAM in the above configurations. However, due to the large feature resolution, decoder computations also come into play, given high-resolution images in the MS-COCO benchmark (Lin et al., 2014). Hence, although we have analyzed its effect (See supplement), we do not use this configuration.

## 4    EXPERIMENTS

**Dataset and Evaluation Metric.** Following (Carion et al., 2020; Liu et al., 2022; Li et al., 2022), we use MS-COCO 2017 benchmark (Lin et al., 2014) for evaluation, having 117k training images and 5k validation images. We use MS-COCO's standard evaluation metric of Average Precision (AP) at different thresholds and different object scales.

**Implementation Details.** We plug the proposed CRED into state-of-the-art DN-DETR (Li et al., 2022) for all our experimental evaluations, including ablations. However, to showcase generality, we also adapt CRED into other prominent DETR methods, e.g., Conditional DETR (Meng et al., 2021), DAB-DETR (Liu et al., 2022) etc.

We perform experiments in the 50-epoch setting, widely used for DETRs (Liu et al., 2022; Li et al., 2022; Zhang et al., 2023a). We also show results for the 12-epoch or $1\times$ (Li et al., 2022) setting to demonstrate accelerated convergence due to the improved DETR design in this paper. The base learning for the backbone is set to $1 \times 10^{-5}$ while for the transformer, it is set to $1 \times 10^{-4}$. For the 12-epoch schedule, the learning rate is dropped by 0.1 at $11^{th}$ epoch, whereas it is dropped at $40^{th}$ epoch for the 50-epoch schedule. We use $8\times$ NVIDIA A40 with a batch size of 16 (2 per-GPU) for training. All the ablations are performed at the $1\times$ setting.

### 4.1    MAIN RESULTS

**CRED in DETR: 50-Epoch Setting.** We plug CRED into representative DETR frameworks. Table 1 shows that CRED boosts the AP in each DETR pipeline. Compared with the baseline without DC, CRED introduces a slight overhead of 9G FLOPs with only $1 - 2$FPS drop. Compared with recent IMFA (Zhang et al., 2023a), the overhead is $\sim 36\%$ less with an improvement of $+4$FPS. Our CRED also performs better regarding runtime than the recent sparse sampling-based method (Zhang et al., 2023a).

Further, when plugged in Conditional-DETR (Meng et al., 2021), CRED delivers the same accuracy at 50% more FPS than the advanced high-resolution DAB-DETR-DC5-R50 (Li et al., 2022). Similarly, with DN-DETR, CRED delivers the same accuracy at 76% more FPS and 50% fewer FLOPs.

**CRED in DETR: 12-Epoch Setting.** Evaluations in this setting show that CRED speeds up convergence with slight overhead. From Table 1, CRED with DAB-DETR (Liu et al., 2022) is better than vanilla DAB-DETR by 3.3AP with only a drop of 2FPS. Compared with high-resolution DAB-DETR-DC5, CRED is accurate by 0.4AP at 50% fewer FLOPs and 76% more FPS.

With DN-DETR (Li et al., 2022), CRED achieves beyond 41AP in just 103G FLOPs, implying that CRED improves the convergence speed (See Figure 5), i.e. DN-DETR via CRED achieves the performance of its DC counterpart in just 12 epochs at 50% fewer FLOPs and 76% higher FPS. From the table, CRED can improve the performance of smaller backbones like ResNet18 while delivering real-time performance ($> 30$FPS). This indicates the utility of CRED where smaller backbones are used due to resource constraints. Hence, by using CRED, detection performance can be boosted

**CRED in DETR: DC$\times$0.25 Setting.** This setting is crucial to show the utility of CRED in DETRs for real-time performance. From Table 1, when encoder resolution is dropped to half (DC$\times$0.25)

Table 1: Comprehensive evaluation of CRED when applied in prominent DETR models under different settings, i.e., training duration, encoder resolution. Our CRED performs better and faster than DETRs operating at high resolution (DC). 'R50: ResNet50 and 'R18: ResNet18 He et al. (2016). Refer to Sec.3.3 for DC×0.25.

| Method | #Epochs | #Params | #FLOPs | #FPS | AP | AP$_{50}$ | AP$_{75}$ | AP$_S$ | AP$_M$ | AP$_L$ |
|---|---|---|---|---|---|---|---|---|---|---|
| Conditional DETR-R50 Meng et al. (2021) | 50 | 44M | 90G | 26 | 40.9 | 61.8 | 43.3 | 20.8 | 44.6 | 59.2 |
| Conditional DETR-DC5-R50 Meng et al. (2021) | 50 | 44M | 195G | 15 | 43.8 | 64.4 | 46.7 | 24.0 | 47.6 | 60.7 |
| • Conditional DETR-R50 Meng et al. (2021) + CRED | 50 | 45M | **100G** | 25 | **44.4** | 64.6 | 47.8 | 25.2 | 46.9 | 60.7 |
| DAB DETR-R50 Liu et al. (2022) | 50 | 44M | 94G | 25 | 42.2 | 63.1 | 44.7 | 21.5 | 45.7 | 60.3 |
| DAB DETR-DC-R50 Liu et al. (2022) | 50 | 45M | 202G | 13 | 44.5 | 65.1 | 47.7 | 25.3 | 48.2 | 62.3 |
| DAB DETR-R50 Liu et al. (2022) + IMFA Zhang et al. (2023a) | 50 | 53M | 108G | 18 | 45.5 | 65.0 | 49.3 | 27.3 | 48.3 | 61.6 |
| • DAB DETR-R50 Liu et al. (2022) + CRED | 50 | 45M | **103G** | 23 | **45.4** | 64.9 | 49.4 | 27.0 | 48.5 | 62.2 |
| DN-DETR-R50 Li et al. (2022) | 50 | 44M | 94G | 25 | 44.1 | 64.4 | 46.7 | 22.9 | 48.0 | 63.4 |
| DN-DETR-DC5-R50 Liu et al. (2022) | 50 | 44M | 202G | 13 | 46.3 | 66.4 | 49.7 | 26.7 | 50.0 | 64.3 |
| • DN-DETR-R50 Li et al. (2022) + CRED | 50 | 45M | **103G** | 23 | **46.2** | 65.8 | 49.8 | 26.8 | 50.0 | 63.5 |
| **12 Epoch Schedule** | | | | | | | | | | |
| Conditional DETR-R50 Meng et al. (2021) | 12 | 44M | 90G | 26 | 32.4 | 52.1 | 33.9 | 14.2 | 35.2 | 48.4 |
| • Conditional DETR-R50 + CRED | 12 | 45M | **100G** | 25 | **36.6** | 56.2 | 38.7 | 18.8 | 39.5 | 52.6 |
| DAB DETR-R50 Liu et al. (2022) | 12 | 44M | 94G | 25 | 35.1 | 55.5 | 36.7 | 16.2 | 38.1 | 52.5 |
| DAB DETR-R50-DC5 Liu et al. (2022) | 12 | 44M | 202G | 13 | 38.0 | 60.3 | 39.8 | 19.2 | 40.9 | 55.4 |
| DAB DETR-R50 Liu et al. (2022) + IMFA Zhang et al. (2023a) | 12 | 53M | 108G | 18 | 37.3 | 57.9 | 39.9 | 20.8 | 40.7 | 52.3 |
| • DAB DETR-R50 Liu et al. (2022) + CRED | 12 | 45M | **103G** | 23 | **38.4** | 58.4 | 41.0 | 20.0 | 41.8 | 53.9 |
| DN-DETR-R50 Li et al. (2022) | 12 | 44M | 94G | 25 | 38.6 | 59.1 | 41.0 | 17.3 | 42.4 | 57.7 |
| DN-DETR-DC5-R50 Li et al. (2022) | 12 | 44M | 202G | 13 | 41.7 | 61.4 | 44.1 | 21.2 | 45.0 | 60.2 |
| • DN-DETR-R50 Li et al. (2022) + CRED | 12 | 45M | **103G** | 23 | **41.1** | 60.6 | 44.0 | 22.2 | 44.1 | 58.9 |
| DAB DETR-R18 Liu et al. (2022) | 12 | 31M | 49G | 38 | 29.8 | 49.0 | 30.5 | 10.9 | 32.5 | 46.9 |
| DAB DETR-R18 Liu et al. (2022) + IMFA Zhang et al. (2023a) | 12 | 40M | 61G | 23 | 31.2 | 51.5 | 32.3 | 13.0 | 33.2 | 49.1 |
| • DAB-DETR-R18 Liu et al. (2022) + CRED | 12 | 32M | **60G** | 31 | **33.5** | 52.0 | 35.2 | 16.7 | 36.0 | 46.2 |
| DN-DETR-R18 Li et al. (2022) | 12 | 31M | 49G | 38 | 32.5 | 51.6 | 33.7 | 13.5 | 35.1 | 49.4 |
| • DN-DETR-R18 Li et al. (2022) + CRED | 12 | 32M | **60G** | 31 | **35.0** | 54.0 | 36.9 | 16.3 | 37.0 | 51.4 |
| **DC×0.25 Configuration** | | | | | | | | | | |
| DAB DETR-R50 Liu et al. (2022) | 12 | 44M | 94G | 25 | 35.1 | 55.5 | 36.7 | 16.2 | 38.1 | 52.5 |
| DAB DETR-R50 Liu et al. (2022) DC×0.25 | 12 | 44M | 80G | 26 | 28.4 | 48.9 | 30.0 | 9.8 | 31.5 | 47.0 |
| DAB DETR-R50 Liu et al. (2022) + IMFA Zhang et al. (2023a) DC×0.25 | 12 | 44M | 96G | 18 | 33.0 | 54.2 | 34.5 | 16.1 | 35.3 | 46.5 |
| • DAB-DETR-R50 Liu et al. (2022) + CRED DC×0.25 | 12 | 45M | **94G** | 24 | **37.5** | 57.9 | 40.1 | 18.8 | 40.7 | 53.0 |
| DN-DETR-R50 Li et al. (2022) | 12 | 44M | 94G | 25 | 38.6 | 59.1 | 41.0 | 17.3 | 42.4 | 57.7 |
| DN-DETR-R50 Li et al. (2022) DC×0.25 | 12 | 44M | 80G | 26 | 31.5 | 52.7 | 31.5 | 10.8 | 33.7 | 52.0 |
| • DN-DETR-R50 Li et al. (2022) + CRED DC×0.25 | 12 | 45M | **94G** | 24 | **40.0** | 59.4 | 42.8 | 20.7 | 43.1 | 56.4 |
| DN-DETR-R50 Li et al. (2022) | 50 | 44M | 94G | 25 | 44.1 | 64.4 | 46.7 | 22.9 | 48.0 | 63.4 |
| DN-DETR-R50 Li et al. (2022) DC×0.25 | 50 | 44M | 80G | 26 | 39.9 | 60.1 | 41.9 | 19.2 | 43.5 | 59.7 |
| • DN-DETR-R50 Li et al. (2022) + CRED DC×0.25 | 50 | 45M | **94G** | 24 | **45.8** | 64.9 | 49.1 | 25.9 | 49.1 | 62.8 |
| DAB DETR-R18 Liu et al. (2022) | 12 | 31M | 49G | 38 | 29.8 | 49.0 | 30.5 | 10.9 | 32.5 | 46.9 |
| DAB DETR-R18 Liu et al. (2022) DC×0.25 | 12 | 31M | 40G | 39 | 24.0 | 43.8 | 25.2 | 4.7 | 27.9 | 42.0 |
| DAB DETR-R18 Liu et al. (2022) + IMFA Zhang et al. (2023a) DC×0.25 | 12 | 40M | 50G | 25 | 27.8 | 46.9 | 28.8 | 14.1 | 29.5 | 29.5 |
| • DAB-DETR-R18 Liu et al. (2022) + CRED DC×0.25 | 12 | 32M | **51G** | 34 | **32.2** | 50.9 | 34.1 | 15.9 | 35.2 | 44.9 |
| DN-DETR-R18 Li et al. (2022) | 12 | 31M | 49G | 38 | 32.5 | 51.6 | 33.7 | 13.5 | 35.1 | 49.4 |
| DN-DETR-R18 Li et al. (2022) DC×0.25 | 12 | 31M | 40G | 39 | 27.0 | 46.4 | 26.9 | 8.3 | 28.0 | 45.6 |
| • DN-DETR-R18 Li et al. (2022) + CRED DC×0.25 | 12 | 32M | **51G** | 34 | **34.2** | 53.0 | 36.2 | 16.0 | 36.1 | 50.0 |

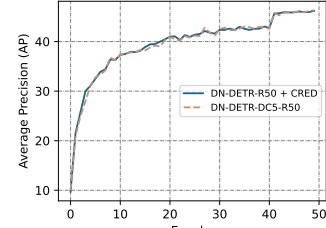

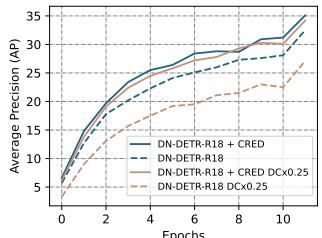

Figure 5: Convergence plots over MS-COCO validation set. (a) It can be seen that despite having 50% fewer FLOPs and 76% higher FPS, CRED converges similarly to the baseline. (b) DETR with smaller backbone and their DC×0.25 variants in 12 epoch setting. Notice that even in the smaller backbone, CRED-enabled model with and without DC×0.25 have similar accuracy, but this gap is noticeable in the baselines with and without DC×0.25. This strengthens the utility of CRED that encoder input resolution can be aggressively dropped to save computations while having better accuracy.

of any vanilla DETR pipeline, it degrades the performance while also reducing FLOP requirement. However, the degradation in the performance supersedes the reduced FLOPs.

Whereas CRED in this configuration delivers performance better than the vanilla variant. For example, for vanilla DAB-DETR-R50, the AP drops from 35.1 to 28.4 with DC×0.25; however, by using CRED in DC×0.25 configuration, we achieve +2.4AP than the vanilla DAB-DETR-R50 in same FLOPs and same FPS. A similar case applies to DN-DETR variants with different backbones.

Table 2: Comparison with state-of-the-art object detectors on COCO val 2017. 'MS': Multiscale, 'SM': Sparse Multiscale Sampling, and 'DC': Dilated Convolution. FPS is reported at $(800 \times 1280)$

| Method | MS | SM | DC | #Epochs | #Params | #FLOPs | #FPS | AP | $AP_{50}$ | $AP_{75}$ | $AP_S$ | $AP_M$ | $AP_L$ |
|---|---|---|---|---|---|---|---|---|---|---|---|---|---|
| YOLOS-DeiT-S Fang et al. (2021) | | | | 150 | 28M | 172G | – | 37.6 | 57.6 | 39.2 | 15.9 | 40.2 | 57.3 |
| Faster-RCNN-FPN-R50 Ren et al. (2015); Lin et al. (2017) | ✓ | | | 108 | 42M | 180G | – | 42.0 | 62.1 | 45.5 | 26.6 | 45.5 | 53.4 |
| TSP-FCOS-FPN-R50 Sun et al. (2021b) | ✓ | | | 36 | 52M | 189G | – | 43.1 | 62.3 | 47.0 | 26.6 | 46.8 | 55.9 |
| TSP-RCNN-FPN-R50 Sun et al. (2021b) | ✓ | | | 36 | 64M | 188G | – | 43.8 | 63.3 | 48.3 | 28.6 | 46.9 | 55.7 |
| Faster RCNN-FPN-R101 Ren et al. (2015); Lin et al. (2017) | ✓ | | | 108 | 60M | 246G | – | 44.0 | 63.9 | 47.8 | 27.2 | 48.1 | 56.0 |
| Sparse-RCNN-FPN-R50 Sun et al. (2021a) | ✓ | | | 36 | 106M | 166G | – | 45.0 | 64.1 | 48.9 | 28.0 | 47.6 | 59.5 |
| DETR-DC5-R50 Carion et al. (2020) | | | ✓ | 500 | 41M | 187G | 16 | 43.3 | 63.1 | 45.9 | 22.5 | 47.3 | 61.1 |
| SAM-DETR-DC5-R50 Zhang et al. (2022) | | | ✓ | 50 | 58M | 210G | – | 43.3 | 64.4 | 46.2 | 25.1 | 46.9 | 61.0 |
| SMCA-DETR-R50 Gao et al. (2021a) | ✓ | | | 50 | 40M | 152G | – | 43.7 | 63.6 | 47.2 | 24.2 | 47.0 | 60.4 |
| Deformable-DETR-R50 Zhu et al. (2021) | ✓ | | | 50 | 40M | 173G | – | 43.8 | 62.6 | 47.7 | 26.4 | 47.1 | 58.0 |
| Conditional-DETR-DC5-R50 Meng et al. (2021) | | | ✓ | 50 | 44M | 195G | 15 | 43.8 | 64.4 | 46.7 | 24.0 | 47.6 | 60.7 |
| Anchor-DETR-DC5-R50 Wang et al. (2021) | | | ✓ | 50 | 37M | 172G | – | 44.2 | 64.7 | 47.5 | 24.7 | 48.2 | 60.6 |
| Efficient-DETR-R50 Yao et al. (2021) | ✓ | | | 36 | 32M | 159G | – | 44.2 | 62.2 | 48.0 | 28.4 | 47.5 | 56.6 |
| DAB-DETR-DC5-R50 Liu et al. (2022) | | | ✓ | 50 | 44M | 202G | 13 | 44.5 | 65.1 | 47.7 | 25.3 | 48.2 | 62.3 |
| SAM-DETR-DC5-R50 Zhang et al. (2022) w/ SMCA | | | ✓ | 50 | 58M | 210G | – | 45.0 | 65.4 | 47.9 | 26.2 | 49.0 | 63.3 |
| Conditional DETR-DC5-R101 Meng et al. (2021) | | | ✓ | 50 | 63M | 262G | 10 | 45.0 | 65.5 | 48.4 | 26.1 | 48.9 | 62.8 |
| DN-DETR-R101 Li et al. (2022) | | | ✓ | 50 | 63M | 174G | – | 45.2 | 65.5 | 48.3 | 24.1 | 49.1 | 65.1 |
| Deformable-DAB-DETR-R50 Zhu et al. (2021) | ✓ | ✓ | | 50 | 41M | 173G | 12 | 45.4 | 64.7 | 49.0 | 26.8 | 48.3 | 61.7 |
| IMFA-DAB-DETR-R50 Zhang et al. (2023a) | | ✓ | | 50 | 53M | 108G | 18 | 45.5 | 65.0 | 49.3 | 27.3 | 48.3 | 61.6 |
| DAB DETR-DC5-R101 Liu et al. (2022) | | | ✓ | 50 | 63M | 282G | 10 | 45.8 | 65.9 | 49.3 | 27.0 | 49.8 | 63.3 |
| SAP-DETR-DC5-R50 Liu et al. (2023) | | | | 50 | 47M | 197G | 12 | 46.0 | 65.5 | 48.9 | 26.4 | 50.2 | 62.6 |
| DN-DETR-DC5-R50 Li et al. (2022) | | | ✓ | 50 | 44M | 202G | 13 | 46.3 | 66.4 | 49.7 | 26.7 | 50.0 | 64.3 |
| Siamese-DETR-R50 Chen et al. (2023) | ✓ | ✓ | | 50 | 41M | 173G | – | 46.3 | 64.6 | 50.5 | 28.1 | 50.1 | 61.5 |
| Lite-DETR-R50 Li et al. (2023) | ✓ | ✓ | | 50 | 41M | 123G | 15 | 46.7 | 66.1 | 50.6 | 29.1 | 49.7 | 62.2 |
| SAP-DETR-DC5-R101 Liu et al. (2023) | | | | 50 | 67M | 266G | 11 | 46.9 | 66.7 | 50.5 | 27.9 | 51.3 | 64.3 |
| • DAB DETR-R50 Liu et al. (2022) + CRED | | | | 50 | 45M | 103G | 23 | 45.4 | 64.9 | 49.4 | 27.0 | 48.5 | 62.2 |
| • DN-DETR-R50 Li et al. (2022) + CRED DC×0.25 | | | | 50 | 45M | **94G** | 24 | 45.8 | 64.9 | 49.1 | 25.9 | 49.1 | 62.8 |
| • DN-DETR-R50 Li et al. (2022) + CRED | | | | 50 | 45M | 103G | 23 | 46.2 | 65.8 | 49.8 | 26.8 | 50.0 | 63.5 |
| • DN-DETR-R50 Li et al. (2022) + CRED-OO | | | | 50 | 45M | 105G | 23 | 46.8 | 66.8 | 50.5 | 27.4 | 50.7 | 64.0 |
| **12 Epoch Schedule** | | | | | | | | | | | | | |
| DETR-R50 Carion et al. (2020) | | | | 12 | 41M | 86G | 27 | 15.5 | 29.4 | 14.5 | 4.3 | 15.1 | 26.7 |
| Deformable DETR-R50 Zhu et al. (2021) | ✓ | | | 12 | 40M | 173G | 12 | 37.2 | 55.5 | 40.5 | 21.1 | 40.7 | 50.5 |
| DAB DETR-R50 Liu et al. (2022) + IMFA Zhang et al. (2023a) | ✓ | | | 12 | 53M | 108G | 18 | 37.3 | 57.9 | 39.9 | 20.8 | 40.7 | 52.3 |
| DAB DETR-DC-R101 Carion et al. (2020) | | | ✓ | 12 | 63M | 282G | 10 | 40.3 | 62.6 | 42.7 | 22.2 | 44.0 | 57.3 |
| • DN-DETR-R18 Li et al. (2022) + CRED DC×0.25 | | | | 12 | 32M | **51G** | 34 | 34.2 | 53.0 | 36.2 | 16.0 | 36.1 | 50.0 |
| • DN-DETR-R18 Li et al. (2022) + CRED | | | | 12 | 32M | **60G** | 31 | 35.0 | 54.0 | 36.9 | 16.3 | 37.0 | 51.4 |
| • DAB-DETR-R50 Liu et al. (2022) + CRED DC×0.25 | | | | 12 | 45M | **94G** | 24 | 37.5 | 57.9 | 40.1 | 18.8 | 40.7 | 53.0 |
| • DAB DETR-R50 Liu et al. (2022) + CRED | | | | 12 | 45M | 103G | 23 | 38.4 | 58.4 | 41.0 | 20.0 | 41.8 | 53.9 |
| • DN-DETR-R50 Li et al. (2022) + CRED DC×0.25 | | | | 12 | 45M | **94G** | 24 | 40.0 | 59.4 | 42.8 | 20.7 | 43.1 | 56.4 |
| • DN-DETR-R50 Li et al. (2022) + CRED | | | | 12 | 45M | 103G | 23 | 41.1 | 60.6 | 44.0 | 22.2 | 44.1 | 58.9 |

**CRED vs State-of-the-art.** We also compare our CRED with state-of-the-art object detection pipelines with multiscale, high-resolution, and sparse sampling approaches. Table 2 shows the results. From the table, it can be seen that CRED-DETR models are better by a large margin ($> 50\%$) in FLOPs and FPS while delivering accuracy comparable with state-of-the-art methods. Even ResNet-18 based models with CRED show competitive performance with Deformable-DETR (Zhu et al., 2021), DAB-DETR (Liu et al., 2022), IMFA (Zhang et al., 2023a) with a stronger backbone ResNet-101.

CRED w/ ResNet-50 performs better than heavy models, even in DC×0.25 configuration and 12-epoch setting. For example, DN-DETR-R50 + CRED DC×0.25 is better than multiscale Deformable-DETR-R50 (Zhu et al., 2021) by 2.8AP, 45% fewer FLOPs and 50% higher FPS. Similarly, DN-DETR-R50 + CRED is better than DAB-DETR-DC5-R101 by 0.8AP, 63% fewer FLOPs, and 130% higher FPS. Then DN-DETR-R50 + CRED-OO has 60% fewer FLOPs than (Liu et al., 2023) with the same accuracy. Furthermore, the accuracy can be improved using the latest DETR-training techniques of (Zhao et al., 2024a; Hou et al., 2024), which we leave for future work.

Figure 6 further strengthens our results, that CRED, while delivering comparable performance to the state-of-the-art, have far fewer FLOPs and higher FPS. Also, the results indicate that by utilizing the DC×0.25 configuration in CRED, DETRs of real-time speed and high accuracy can be constructed, indicating the huge potential of Cross-Resolution Encoding-Decoding in state-of-the-art DETRs.

## 4.2 ABLATIONS

We conduct a comprehensive ablation study on CRED design by using the state-of-the-art DN-DETR (Li et al., 2022) framework and provide insight on the design motivations.

**Effect of CRAM and OSMA.** We analyze the effect of OSMA (Sec. 3.2) and CRAM (Sec. 3.1). Table 3 shows the analysis. It can be seen that by using any of OSMA or CRAM into the baseline, the accuracy improves. By using OSMA alone, AP increases by 1.2, indicating that OSMA produces better input features or tokens for the encoder. While by using only CRAM, AP improves by 1.8AP.

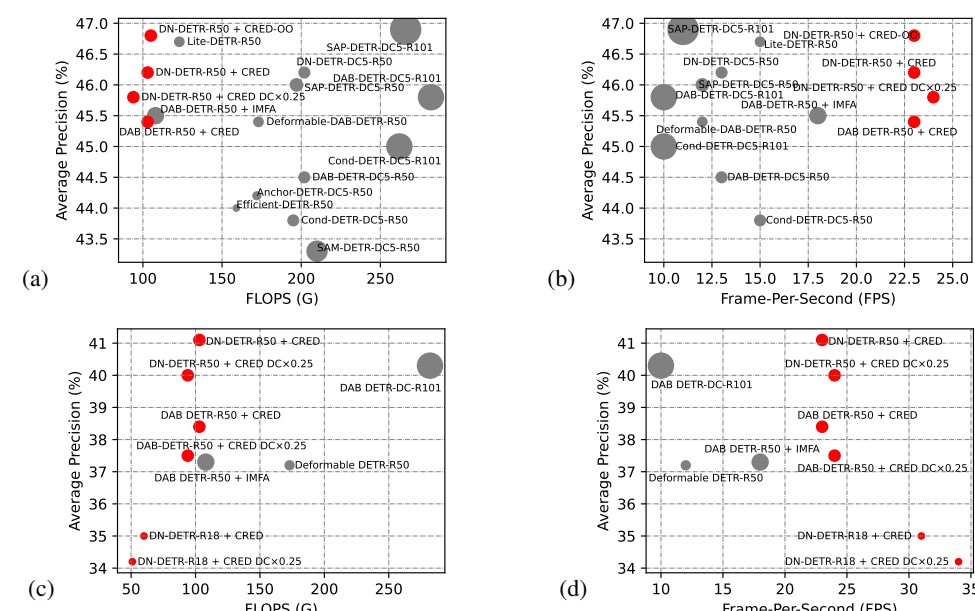

(a)  (b)  (c)  (d)

Figure 6: The proposed CRED *vs* representative DETRs Liu et al. (2022); Li et al. (2022); Zhu et al. (2021); Liu et al. (2023); Li et al. (2023) etc. (a,b) 50-Epochs, and (c,d) 12-epochs. '●' and '●' refers to CRED and existing models respectively. The size of the circle is proportional to the parameter count.

Table 3: Effect of OSMA and CRAM in CRED.

| OSMA | CRAM | #Params | #FLOPs | AP | $AP_{50}$ | $AP_{75}$ | $AP_S$ | $AP_M$ | $AP_L$ |
|------|------|---------|--------|------|------|------|------|------|------|
|      |      | 44M | 94G | 38.6 | 59.1 | 41.0 | 17.3 | 42.4 | 57.7 |
| ✓    |      | 45M | 97G | 39.8 | 59.7 | 42.5 | 19.1 | 43.2 | 58.1 |
|      | ✓    | 45M | 100G | 40.4 | 60.1 | 43.1 | 21.2 | 43.6 | 58.3 |
| ✓    | ✓    | 45M | 103G | 41.1 | 60.6 | 44.0 | 22.2 | 44.1 | 58.9 |

Table 4: Computation complexity analysis of CRED.

| Method | Backbone | Encoder | Decoder | CRAM | OSMA | Total | FPS | $AP_{50}$ |
|--------|----------|---------|---------|------|------|-------|-----|------|
| DN-DETR-R50 Li et al. (2022) | 74G | 12G | 8G | – | – | 94G | 25 | 44.1 |
| DN-DETR-DC5-R50 Li et al. (2022) | 112G | 80G | 10G | – | – | 202G | 13 | 46.3 |
| ● DN-DETR-R50 Li et al. (2022) + CRED | 74G | 12G | 10G | 3G | 2G | 103G | 23 | 46.2 |

Using both the modules, we get an overall improvement of 2.5AP at a reduction of only 2FPS and negligible parameter overhead. Interestingly, AP for small objects increases by $\sim$ 5AP. If we compare this with the DC variant of the baseline, we perform on par in roughly 99G fewer FLOPs, which is inspirational. This accuracy gap is reduced in the 50 epoch setting.

Hence, we conjecture that feeding multiscale information to the encoder via OSMA while transferring the low-resolution encoded information to high-resolution feature via CRAM proves to be highly effective in DETRs from both accuracy and runtime perspective.

**Computational Complexity.** Table 4 shows the effect of using CRAM and OSMA in DN-DETR. It can be seen that the overhead of these modules is $\sim$ 5G FLOPs. However, overall computations slightly increase due to the increased resolution of the decoder when fed by CRAM.

**Ablation of OSMA Hyperparameters.** OSMA mainly has intermediate projection dimension $d$ and grid size $g$ as hyperparameters. Another ablation exists within the OSMA design, i.e., the number of input multiscale features. We study them individually in Table 5.

The first two rows show the effect of input multiscale features. As we include more high-resolution features, overall AP increases along with the AP of small objects. This indicates OSMA's one-step attention mechanism effectively produces feature maps infused with multiscale information.

We also change grid size $g$. Increasing the grid size $g_0 = 2$ reduces the FLOPs by 2G because $T$ (Figure 3) increases and $N_g$ decreases. However, we observed a reduction in the AP. We hypothesize that this happens because the stage-5 ($F^5$) feature is the smallest resolution. When more than two

Table 5: Ablation of OSMA design.

| ablation | $F^5$ | $F^4$ | $F^3$ | $g_0$ | $P$ | $d$ | #Params | #FLOPs | AP | $AP_{50}$ | $AP_{75}$ | $AP_S$ | $AP_M$ | $AP_L$ |
|---|---|---|---|---|---|---|---|---|---|---|---|---|---|---|
| changing | ✓ | ✓ | | 1 | 1 | 21 | 45M | 101G | 40.2 | 59.5 | 43.2 | 21.3 | 43.2 | 57.9 |
| #scales | ✓ | ✓ | ✓ | 1 | 1 | 21 | 45M | 103G | 41.1 | 60.6 | 44.0 | 22.2 | 44.1 | 58.9 |
| Vary '$\{g_0,P\}$' | ✓ | ✓ | ✓ | 2 | 2 | 21 | 45M | 101G | 39.7 | 59.5 | 42.3 | 21.0 | 42.7 | 56.8 |
| Same res. | ✓ | ✓ | ✓ | 1 | 1 | 21 | 45M | 103G | 41.1 | 60.6 | 44.0 | 22.2 | 44.1 | 58.9 |
| Vary '$\{g_0,P\}$' | ✓ | ✓ | ✓ | 2 | 1 | 21 | 45M | 94G | 40.0 | 59.4 | 42.8 | 20.7 | 43.1 | 56.4 |
| DC×0.25. | ✓ | ✓ | ✓ | 4 | 4 | 21 | 45M | 92G | 39.4 | 59.0 | 42.1 | 19.8 | 42.7 | 56.0 |
| Vary '$d$' | ✓ | ✓ | ✓ | 1 | 1 | 40 | 45M | 114G | 41.5 | 61.1 | 44.2 | 22.4 | 44.8 | 59.6 |

features using $g_0 = 2$ are fused with high-res features, the individual feature at low-resolution loses its chance to interact with the high-resolution features individually because these features already have relatively large receptive fields and carry more information.

Although we are interested in keeping the values of $d$ equal to $T$, we analyze its effect. We observe that it increases the FLOPs while slightly improving the AP. Hence, based on the computational budget requirements, one can change $d$ to achieve the desired performance and runtime.

Table 6: Effect of LayerNorm Ba et al. (2016) and activations in CRED.

| LayerNorm | Activation | #Params | #FLOPs | AP | $AP_{50}$ | $AP_{75}$ | $AP_S$ | $AP_M$ | $AP_L$ |
|---|---|---|---|---|---|---|---|---|---|
| ✓ | ReLU | 45M | 103G | 40.6 | 60.4 | 43.6 | 21.6 | 43.6 | 57.9 |
| ✓ | SiLU | 45M | 103G | 41.1 | 60.6 | 44.0 | 22.2 | 44.1 | 58.9 |
| ✗ | SiLU | 45M | 103G | 39.3 | 59.1 | 41.9 | 20.8 | 42.5 | 56.6 |

Table 7: Ablation of CRAM design.

| Input | #Params | #FLOPs | AP | $AP_{50}$ | $AP_{75}$ | $AP_S$ | $AP_M$ | $AP_L$ |
|---|---|---|---|---|---|---|---|---|
| $F^4$ | 45M | 103G | 41.1 | 60.6 | 44.0 | 22.2 | 44.1 | 58.9 |
| $F^3$ | 45M | 147G | 42.1 | 61.4 | 44.9 | 23.3 | 44.8 | 59.4 |

**CRED Design.** Within the CRED design, we study the effect of different activations and the specified use of layer normalization (Ba et al., 2016). Table 6 shows that using ReLU or removing LayerNorm from CRED decreases accuracy. This justifies the configuration described in the paper.

**Ablation of CRAM.** CRAM is studied by changing its input resolution and source. Table 7 shows that despite feeding the encoder with low resolution, CRAM can effectively transfer the encoder knowledge to the high-resolution feature. By default, we feed resolution equal to $F^4$ to CRAM. When we feed CRAM with $F^3$, the computations in the decoder increase mainly in the cross-attention. Although it improves AP, the rise in FLOPs is notable. Hence, we restrict ourselves to feeding CRAM with resolution up to $F^4$.

## 5 CONCLUSION

In this work, we present a novel Cross-Resolution Encoding-Decoding (CRED) mechanism to improve the accuracy and runtime of DETR methods. CRED is based on its two novel modules Cross Resolution Attention Module (CRAM) and One Step Multiscale Attention (OSMA). CRAM transfers the knowledge of low-resolution encoder output to a high-resolution feature. While OSMA is designed to fuse multiscale features in a single step and produce a feature map of a desired resolution. With the application of CRED into state-of-the-art DETR methods, FLOPs get reduced by 50%, and FPS increases by 76% than the high-resolution DETR at equivalent detection performance.

**Future Scope & Limitations:** CRED with its promising results shows huge potential in real-time and affordable DETRs with high accuracy and high-resolution image processing. There is greater scope for improvements, e.g., fusing CRAM and OSMA for even higher performance or adapting CRED to sparse sampling-based DETRs because the current design can not fuse high-resolution features with sparsely sampled encoder embeddings. In addition, CRED has huge scope in Transformer-based semantic or instance segmentation by leveraging its attention transfer to improve runtime for processing high-resolution images because semantic segmentation produces high-resolution outputs.

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

DETECTION VISUALIZATIONS ON MS-COCO VALIDATION SET

# A QUALITATIVE COMPARISON WITH DN-DETR-R50 (LI ET AL., 2022) AND DN-DETR-DC5-R50 LI ET AL. (2022)

To put the qualitative results in context, we have compared CRED-DETR with baseline DN-DETR with low resolution and DC5 setting. Please refer to Figure 7.

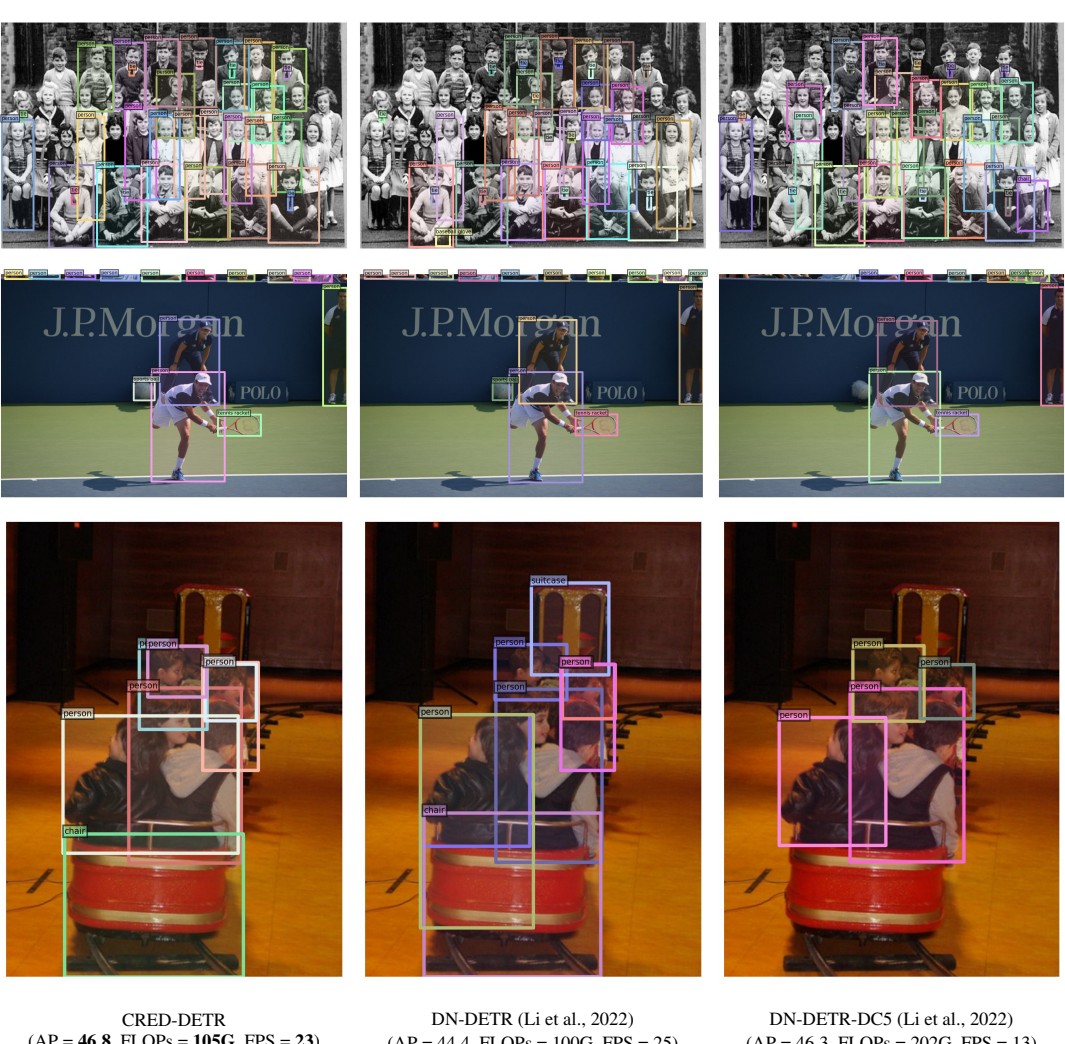

|  |  |  |
|---|---|---|
| CRED-DETR | DN-DETR (Li et al., 2022) | DN-DETR-DC5 (Li et al., 2022) |
| (AP = **46**.**8**, FLOPs = **105**G, FPS = **23**) | (AP = 44.4, FLOPs = 100G, FPS = 25) | (AP = 46.3, FLOPs = 202G, FPS = 13) |

Figure 7: It can be seen that the proposed CRED-DETR can detect objects with high accuracy. *Top*: CRED-DETR detects almost all of the persons standing in the top (4th row) in the image, including the tie. In contrast, baselines struggle to achieve the same as indicated in rows 1-3 in the top image. *Middle*: CRED-DETR can detect the instances of person visible with very small field of view. However, the DC5 variant of the baseline can not be detected. *bottom*: CRED-DETR detects more number of persons. The baseline DN-DETR misclassifies the front region of the train as a suitcase, while CRED-DETR does not. The same is verified with empirical results provided in the paper.

## B  QUALITATIVE ANALYSIS

Below we have visualized detection results from CRED-DETR from MS-COCO 2017 validation set. The images covers wide range of object from tiny, small to large. It can be seen that CRED-DETR, despite running at low resolution in the encoder, is able to detect all categories of object.

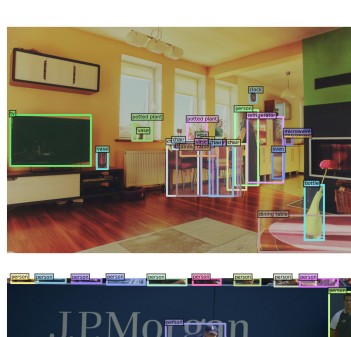 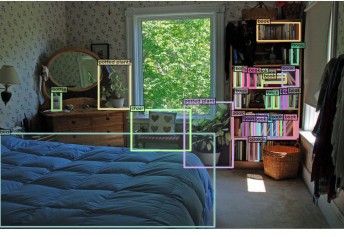 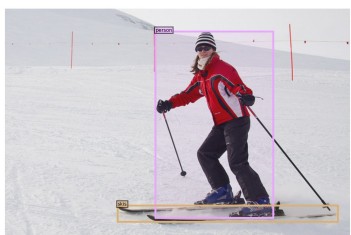

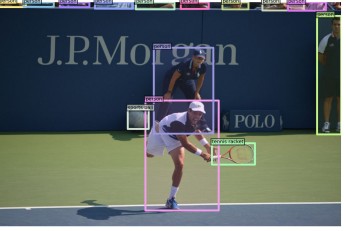 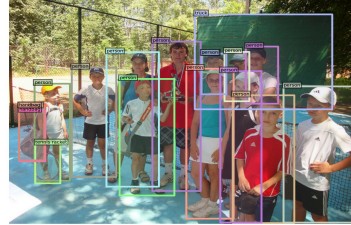 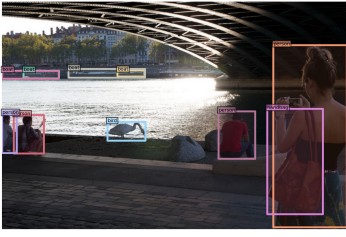

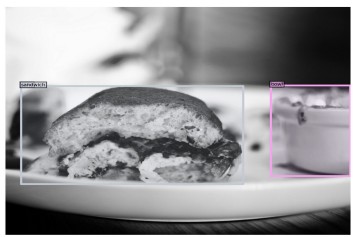 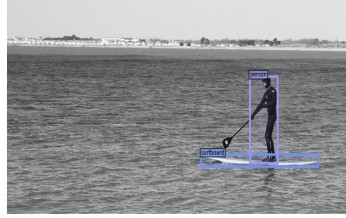 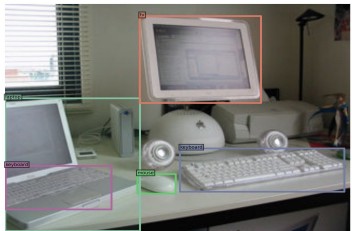

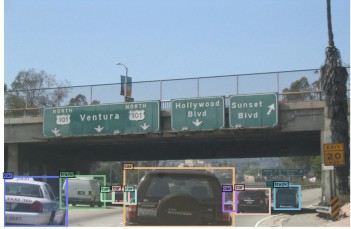 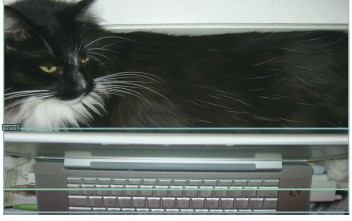 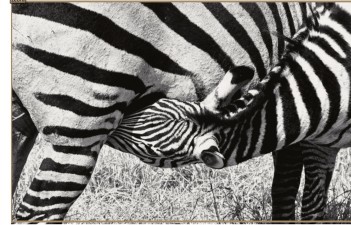

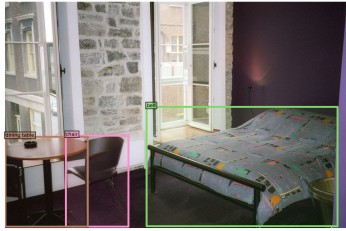 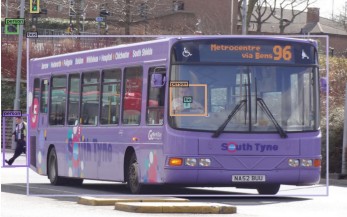 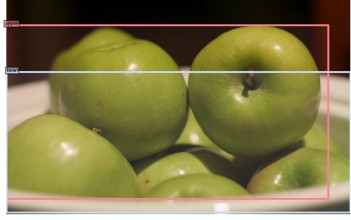

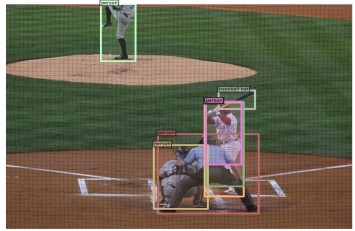 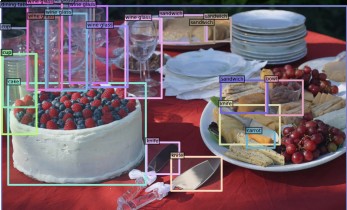 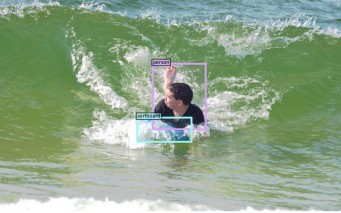

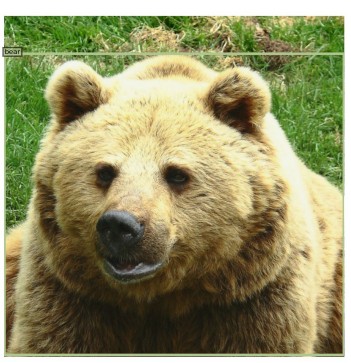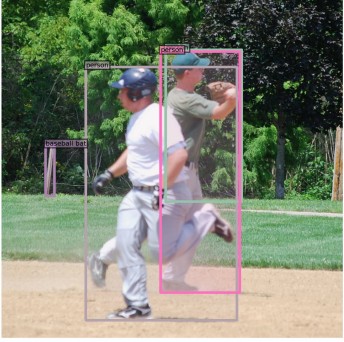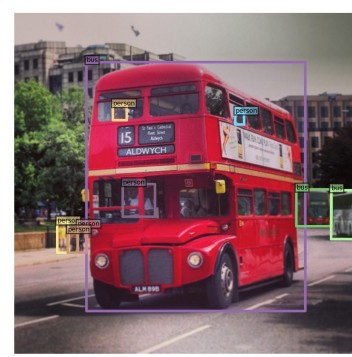

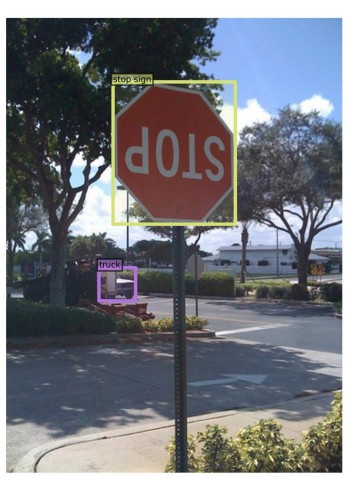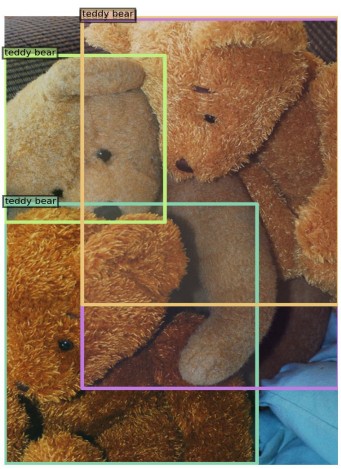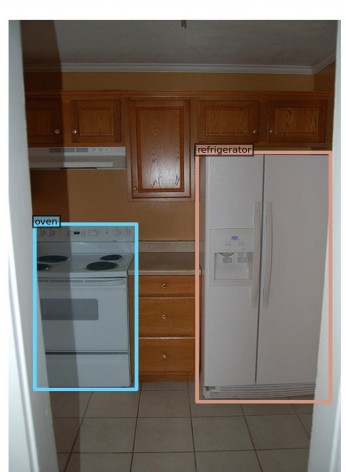

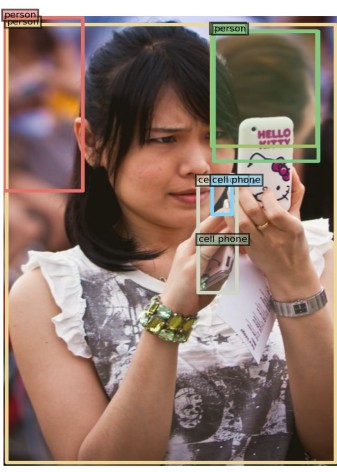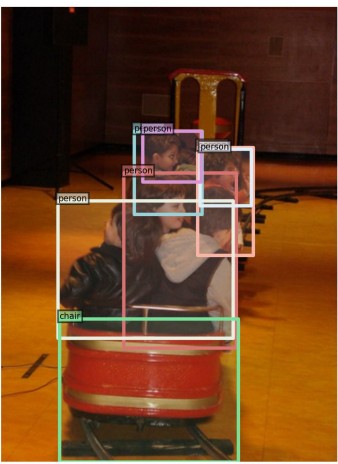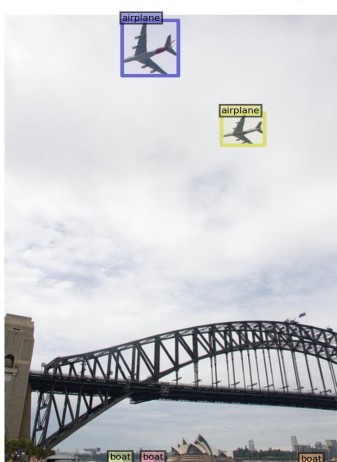

