# OpenReview forum: "Cross Resolution Encoding-Decoding For Detection Transformers"
_ICLR.cc/2025/Conference — Submitted to ICLR 2025_

### Official Review · Reviewer_9nhT · 2024-10-28

**Soundness:** 3
**Presentation:** 3
**Contribution:** 2
**Rating:** 6
**Confidence:** 4

**Summary:**

This paper present a novel method named Cross-Resolution Encoding-Decoding (CRED), which can reduce the computational cost of the traditional DETR model , especially for high-resolution detection. By using CRAM module and OSAM module, the CRED effectively combines low-level and high-level feature to achieve the balance between detection accuracy and computational efficiency. The authors validate the effectiveness of CRED through a series of experiments and ablation studies.

**Strengths:**

The motivation of the paper is well-founded. The high computational burden in high-resolution detection is an important problem, and the combination of both low-level and high-level feature is a reasonable approach.

The experiment results show that the proposed CRED approach improves performance compared to the original DETR, which demonstrates the effectiveness.

**Weaknesses:**

Compare with original DN-DETR，the improvement of CRED-DERT is not significant. More comparative analysis and deeper insights into the benefits of CRED over DN-DETR will strengthen the contribution.

The motivation of introducing multi-scale features is reasonable, but the CRAM has a similar structure to FPN, which seems to be more of an adaptation rather than a breakthrough

Lack of the comparation with SoTA method, such as RT-DETR. [1]. As far as I know, RT-DETR achieves higher throughput and is designed for real-time application. The absence of comparisons with such methods weakens the evaluation of CRED’s performance, particularly in terms of speed and efficiency, which are key claims of the paper.

Lack of visual comparison with other SoTA methods, especially the detecting result of small objects, which is the evidence of the advantage of CRED.

[1] DETRs Beat YOLOs on Real-time Object Detection, CVPR, 2024

**Questions:**

Is the local aggregation of multiscale features module in OSMA really efficient in practice and friendly to the hardware? This module requires a lot of memory discontinuous operations, which may limit the actual running speed of the model.

As shown in Figure1, the input of CRAM module is the feature from Stage-4 or Stage-3, why we do not use Stage-2 or combine the feature from Stage-4 and Stage-3?

---

### Official Review · Reviewer_f5Je · 2024-10-30

**Soundness:** 3
**Presentation:** 2
**Contribution:** 2
**Rating:** 6
**Confidence:** 3

**Summary:**

The paper proposes a new Cross-Resolution Encoding-Decoding (CRED) mechanism aimed at improving the computational efficiency of Detection Transformers (DETR) while maintaining high detection accuracy. The CRED mechanism consists of two key modules: the Cross-Resolution Attention Module (CRAM) and the One Step Multiscale Attention (OSMA). CRAM transfers knowledge from low-resolution encoder outputs to higher-resolution decoder inputs, while OSMA fuses multiscale features in a single step, generating feature maps enriched with multiscale information. When applied to state-of-the-art DETR variants, the proposed method significantly reduces computational complexity, with the authors claiming a 50% reduction in FLOPs and a 76% increase in FPS, all without sacrificing detection accuracy. The paper presents extensive experiments, ablation studies, and comparisons with existing methods to support the claims.

**Strengths:**

1. The CRED mechanism is a sound approach to reduce the computational cost of DETR models by balancing low-resolution encoding with high-resolution decoding.
2. Extensive experiments are implmented of the CRED-enhanced DETR models on the MS-COCO 2017 benchmark. The results show large improvements in FLOPs and runtime without loss in detection accuracy.
3. The paper provides detailed ablation studies, analyzing the contributions of the individual CRAM and OSMA modules. This shows the influence of each component and provides insight into the design choices.
4. The CRED mechanism is shown to be effective across different DETR variants and backbone architectures, indicating that it has a good scalability and can be applied to various object detection pipelines.

**Weaknesses:**

1. The overall design is more like an engineering design and lacks technical novelty. While the empirical performance improvements are well-documented, the paper could benefit from a more in-depth theoretical analysis of the CRED mechanism. For example, why does transferring low-resolution encoder information to high-resolution decoder inputs improve performance?
2. While the authors claim that CRED improves the detection of small objects, the improvements in average precision for small objects (APS) are relatively modest in some configurations. This suggests that while the CRED mechanism provides overall efficiency gains, its impact on small object detection could be further optimized.
3. Although the authors compare CRED against several state-of-the-art DETR models (such as Deformable DETR, DN-DETR, and IMFA), there is a lack of comparison with some of the latest advancements in efficient transformer architectures, such as sparse transformers and vision transformers (ViTs).
4. The paper provides limited qualitative analysis through visualizations of detection results (Figures 7–10). While these figures are informative, a more detailed qualitative analysis of how CRED affects detection in challenging cases (e.g., cluttered backgrounds, occlusions) would strengthen the paper's narrative.

**Questions:**

1. Including the comparison of latest efficient transformers would provide a more comprehensive evaluation of CRED’s competitiveness.
2. Given the modest gains in small object detection (APS), the authors could explore further optimizations to make CRED more effective for this class of objects.

---

### Official Review · Reviewer_5mdu · 2024-11-03

**Soundness:** 4
**Presentation:** 4
**Contribution:** 3
**Rating:** 6
**Confidence:** 4

**Summary:**

This paper proposes a cross-resolution encoding-decoding mechanism to improve the efficiency of DETR. Two important components CRAM and OSMA are introduced to convert the knowledge of the low-resolution encoder output into high-resolution features. Extensive analyses and experiments have demonstrated the effectiveness of the method.

**Strengths:**

1. CRED improves the computational efficiency of DETR by allowing for the accuracy of high-resolution detection while operating at the speed of low-resolution processes.

2. Enhance the DETR block CRAM and OSMA are two main module for CRED. CRAM can facilitate the info transfer between high resolution and low resolution. OSMA simplifying the integration of detailed features across different scales.

3. The ablation experiment is comprehensive and adequate.

**Weaknesses:**

1. More datasets is better, only on MS-COCO2017 didn't quite convince me.

2. While FLOPs and FPS metrics are provided, a more in-depth discussion of the tradeoffs between accuracy, computational cost, and speed of inference would be beneficial a lot.

3. Why is augmenting global information to high-resolution features better than sparse sampling methods(e.g.,IFMA)?

4. The main experiments have used DETR as a baseline, and I would like to know how the method performs on other more advanced detection algorithms, which would be valuable in understanding the generalizability of the method.

**Questions:**

Please see weaknesses.

---

### Official Review · Reviewer_QqZN · 2024-11-03

**Soundness:** 3
**Presentation:** 3
**Contribution:** 3
**Rating:** 5
**Confidence:** 3

**Summary:**

This paper introduces a Cross-Resolution Encoding-Decoding (CRED) mechanism to enhance the efficiency of DETR for object detection. CRED consists of two key modules: the Cross Resolution Attention Module (CRAM) and One Step Multiscale Attention (OSMA). CRAM transfers knowledge from low-resolution outputs to high-resolution features, while OSMA fuses multiscale features in one step. By integrating CRED into DETR models, such as DN-DETR, the approach achieves similar accuracy to high-resolution DETR with fewer FLOPs and faster performance.

**Strengths:**

The proposed approach feeds the encoder with low-resolution features while supplying the decoder with high-resolution features from the backbone. This method achieves an effective speed-accuracy tradeoff. The results are demonstrated using MS-COCO dataset.

**Weaknesses:**

The idea of combining low-resolution and high-resolution features or using multiscale features to enhance DETR performance is not new, as similar approaches have been used in previous studies, such as Zhang et al., (2023a); Zhao et al., (2024b); and Li et al., (2023). How do you justify the novelty of your contributions?

The proposed approach has only been evaluated on a single dataset, MS COCO. While it outperforms the baselines, as shown in Table 1, it achieves only competitive results compared to state-of-the-art methods in Table 2. It would be more convincing if the approach were tested on additional datasets.

**Questions:**

The proposed OSMA combines multiscale features along the channel dimension. However, multiscale features typically have different numbers of channels—high-resolution features usually have fewer channels than low-resolution ones. How did you handle this discrepancy when aggregating them by channel? Did you reduce the number of channels for the low-resolution features? Could you provide more details about the implementation?

Additionally, the citation format needs adjustment. For instance, in some places, citations like "Zhang et al. (2023a)" should be written as “(Zhang et al. 2023a).”

---

### Meta-Review · Area_Chair_K6Xf · 2024-12-20

**Metareview:**

To improve the efficiency of DETR computation, the Cross-Resolution Encoding-Decoding (CRED) mechanism has been proposed. CRED uses two modules, CRAM and OSMA, to achieve both low-resolution speed and high-resolution accuracy. CRAM transfers low-resolution information to high-resolution, and OSMA integrates multi-scale information to improve the desired resolution.

The strengths of this work are as follows. CRED improves the computational efficiency of DETR by achieving the accuracy of high-resolution detection while operating at the speed of low-resolution processes.

The weaknesses of this work are as follows. Although the AC acknowledges the novelty of transferring low-resolution attention computed from low-resolution features to high-resolution features, the general idea of combining low-resolution and high-resolution features is not new, and it is difficult to say that it is incredibly novel. The proposed approach has only been evaluated on a single dataset, MS COCO. Additional experiments have been submitted, but they are not comprehensive.

This paper was evaluated as borderline. As mentioned above, the paper proposes a method to improve the computational efficiency of DETR by achieving the accuracy of high resolution detection while operating at the speed of low resolution processes, and the effectiveness of the method has been recognized. However, in terms of the innovativeness of the method and the comprehensiveness of the experiments, it is difficult to say that it is a strong factor in overcoming the threshold, and at this point it is judged to be rejected. The AC recommends that the authors consider the reviewers' comments and submit the paper to the next conference.

**Additional Comments On Reviewer Discussion:**

The novelty of the method and the comprehensiveness of the The novelty of the method and the comprehensiveness of the experiments were the main points of discussion in this paper. Certainly, as the authors claim, the novelty of the object detection method that transfers attention from low to high resolution is recognized. On the other hand, it is questionable whether the proposed idea is innovative within the larger framework of combining high- and low-resolution features.

In addition, regarding the comprehensiveness of the experiments, the authors have added the PASCAL-VOC-2012 experiment in addition to the MS-COCO-2017 experiment. While the AC acknowledges the efforts of the authors, the AC has some concerns about whether the additional experiment is comprehensive.

---

### Decision · Program_Chairs · 2025-01-22

Reject